# WGCNA Analysis Revealed the Hub Genes Related to Soil Cadmium Stress in Maize Kernel (*Zea mays* L.)

**DOI:** 10.3390/genes13112130

**Published:** 2022-11-16

**Authors:** Yongjin Li, Ying Zhang, Hongbing Luo, Dan Lv, Zhenxie Yi, Meijuan Duan, Min Deng

**Affiliations:** 1College of Agronomy, Hunan Agricultural University, Changsha 410128, China; 2College of Agronomy, Northwest A&F University, Xianyang 712100, China; 3Maize Engineering Technology Research Center of Hunan Province, Changsha 410128, China

**Keywords:** maize, Cd stress, WGCNA, gene co-expression, network

## Abstract

Soil contamination by heavy metals has become a prevalent topic due to their widespread release from industry, agriculture, and other human activities. Great progress has been made in elucidating the uptake and translocation of cadmium (Cd) accumulation in rice. However, there is still little known about corresponding progress in maize. In the current study, we performed a comparative RNA-Seq-based approach to identify differentially expressed genes (DEGs) of maize immature kernel related to Cd stress. In total, 55, 92, 22, and 542 DEGs responsive to high cadmium concentration soil were identified between XNY22-CHS-8 vs. XNY22-YA-8, XNY22-CHS-24 vs. XNY22-YA-24, XNY27-CHS-8 vs. XNY27-YA-8, and XNY27-CHS-24 vs. XNY27-YA-24, respectively. The weighted gene co-expression network analysis (WGCNA) categorized the 9599 Cd stress-responsive hub genes into 37 different gene network modules. Combining the hub genes and DEGs, we obtained 71 candidate genes. Gene Ontology (GO) enrichment analysis of genes in the greenyellow module in XNY27-YA-24 and connectivity genes of these 71 candidate hub genes showed that the responses to metal ion, inorganic substance, abiotic stimulus, hydrogen peroxide, oxidative stress, stimulus, and other processes were enrichment. Moreover, five candidate genes that were responsive to Cd stress in maize kernel were detected. These results provided the putative key genes and pathways to response to Cd stress in maize kernel, and a useful dataset for unraveling the underlying mechanism of Cd accumulation in maize kernel.

## 1. Introduction

Soil contamination by heavy metals has become a prevalent topic due to their widespread release from industry, agriculture, and other human activities [1]. Cd is one of the most hazardous heavy metals, and it is a pollutant for humans and animals [2]. Because of rapid industrialization and environmental pollution, approximately 20 million hectares of cultivated lands in China, accounting for 20% in total cultivated area, were contaminated with Cd and Pb [3]. Excessive accumulation of Cd in soil could damage crop growth [4] and can cause threat to human health via the plants’ food, such as grains of cereals [5,6].

In theory, strategies to prevent toxic heavy metals entering into plant tissues, and thus into the food chain, can be divided into two types: indirect and direct [7]. One strategy is to decrease Cd availability in soil, such as cleanup of cultivated soils, irrigation water, and air by physical, chemical, and biological methods [7,8,9,10,11,12]. The other strategy is to remove the Cd from soil by planting hyperaccumulators [13,14,15,16]. However, due to their high cost and low efficiency, these strategies are impractical. Therefore, breeding new cultivars with low tissue Cd concentrations is the only viable approach.

Cd accumulation varies widely between plant species. Rice accumulates more Cd than maize, wheat, barley, and sorghum [17]. Most studies have been performed in rice of Cd uptake and translocation, and several related genes were cloned. OsNramp1 and OsNramp5 transports Cd from the soil solution to the root cells of rice, while knockout of them results in complete loss of Cd uptake [18,19]. The vacuolar sequestration of Cd is mediated by *OsHMA3* [20]. Then, the Cd is transported from root cells to grains by transporters *OsHMA2, OsLCT1*, *OsZIP7*, *OsCCX2,* and *OsCd1* [21,22,23,24,25]. Although great progress has been made in elucidating the uptake and translocation of Cd accumulation in rice, little is known about corresponding progress in maize, barley, wheat, and other crops during the last decade.

With the development of the omics technologies, more and more studies have been performed by using them to dissect the molecular mechanism of complex traits. Nowadays, transcriptomics is one of the most common strategies used to dissect the regulation mechanism of biotic and abiotic stress [26,27,28], development [29,30,31], complex traits [32,33], and metabolites biosynthesis [34,35,36]. Recently, more and more transcriptome studies have been performed to screen out a series of candidate genes involved in the responses to Cd stress in various rice [37,38,39], *Brassica juncea* [28], tomato [40], nicotiana [41], wheat [42], cotton [43], barley [44], *Brassica campestris* [45], goldenrain tree [46], and Kenaf [47].

Linkage mapping and GWAS have become effective strategies to reveal the genetic basis of complex agronomic traits. However, few studies have been performed on the genetic architecture of the traits related to Cd stress [48,49,50,51]. Meanwhile, several transcriptomics studies about maize roots response to Cd stress have been already published [52,53,54,55,56]. Therefore, more investigations are needed to determine the effects of Cd accumulation in maize different tissues.

Maize (*Z. mays* L.) is one of the most widely grown cereal crops worldwide. It is an important staple food for people and animals in Africa and South America, and it makes a critical contribution to world food security [57]. Cd accumulation is controlled by multiple genes. Although the molecular mechanism of Cd accumulation in rice and Arabidopsis has been thoroughly studied, only *ZmHMA2*, *ZmHMA3* [49,51,58], *ZmUBP15*, *ZmUBP16,* and *ZmUBP19* [59], which were the homologous genes of *OsHMA2*, *OsHMA3,* and *AtUBP16*, were cloned in maize, respectively. Therefore, exploring the underlying the effects of Cd accumulation in maize, especially in maize kernel, has an important role in breeding maize varieties with low Cd content in maize kernel.

WGCNA is a system biology method which was used to investigate gene association patterns between different samples, and it is an effective method to detect co-expressed modules and hub genes, analyze the relationship between modules and sample phenotypes, and identify key regulatory genes [60,61]. It has been widely used to detect the co-expressed genes responsive to stress in maize [60], Arabidopsis [62,63], and rice [61,64].

In order to investigate the underlying the effects of the regulation networks of immature maize kernel in Cd-stressed soil, two hybrid maize varieties, XNY22 (Xiangnongyu 22) with high Cd concentration in kernel, and XNY27 (Xiangnongyu 27) with low Cd concentration in kernel, were planted in different Cd content soils. The Cd concentration of kernels, roots, and stems 8 days after pollination (DAP), 24 DAP, and 40 DAP of two maize hybrids were measured, respectively. The transcriptome information of the two varieties at two time points was obtained using RNA-Seq. Subsequently, WGCNA analysis was performed to explore the highly correlated modules and co-expressed genes. By analyzing the interaction network within the co-expressed genes of the two varieties, we identified the hub genes and preliminarily explain the different responses of two varieties to Cd stress. Moreover, these results could provide the basis for Cd tolerance improvement by advanced breeding technology in maize.

## 2. Materials and Methods

### 2.1. Plant Materials and Experiment Design

Maize hybrids XNY22 and XNY27 with high and low Cd level accumulation in kernel were used in this study, respectively. These two maize varieties were bred by Hunan Agricultural University. XNY22 and XNY27 were grown in Changsha (28°10′59.08″ N, 113°04′34.22″ E) with low Cd content (0.42 mg/kg) in soil as control and Yong’an (28°15′44.26″ N, 113°20′17.50″ E) with high Cd content (2.73 mg/kg) in soil in March 2018. Hybrids were planted adjacent in a two-row plot with two replicates using a randomized complete block design, and each plot had 20 plants in 2.5 m rows that were separated by 60 cm.

### 2.2. Cd Determination

In this study, the kernels, leaves, roots, and stems 8 DAP, 24 DAP, and 40 DAP from two hybrids were used. After being dried, these samples were ground into powder. The ground material was passed through a 100-mesh sieve. A total of 0.1–0.5 g sample powder was collected and digested with HNO_3_/HClO_4_ (9:1, *v/v*). Then, the Cd concentration of the digestion solution was determined using the ZEEnit700P atomic absorption spectrometer (Analytikjena, Jena, Germany) (Tang et al., 2021).

### 2.3. Samples Collection and RNA Isolation

In this study, 24 immature kernel samples were collected from XNY22 and XNY27, which were planted in high (Yong’an, YA) and low (Changsha, CHS) soil Cd content at 8 DAP and 24 DAP with three biological replicates, respectively. Total RNA was extracted using Trizol reagent according to the manufacturer’s instructions, and mRNA was enriched by Oligo (dT) beads. RNA integrity was assessed using the RNA Nano 6000 Assay Kit of the Agilent Bioanalyzer 2100 system (Agilent Technologies, Santa Clara, CA, USA).

The enriched mRNA was fragmented and reverse-transcribed into cDNA with random primers. Second-strand cDNAs were synthesized by DNA polymerase I, RNase H, dNTP, and buffer. The cDNA fragments were purified using a QiaQuick PCR extraction kit, end-repaired, poly (A)-supplemented, and ligated to Illumina sequencing adapters. RNA library sequencing was performed using an Illumina HiSeqTM2500/4000 by allwegene Biotechnology Co. (Beijing, China).

### 2.4. RNA-Seq Data Processing and Analysis

Raw reads were filtered by removing reads with adaptor sequences, reads in which the percentage of unknown bases (N) was greater than 10%, and low-quality reads (Q < 20) content was greater than 50% [65]. Clean reads were used for mapping, calculation, and normalization of gene expression. All the clean reads in each sample were mapped to B73 (RefGen_V4) genomic DNA sequence using Tophat2 (v2.1.0) [66]. For the DEGs of the immature kernels of two hybrid maize during different developmental stages, the criteria of padj (*p* value was adjusted) < 0.05 was used. Differential expression analysis of two samples was performed using the DESeq (1.10.1) [67].

### 2.5. WGCNA

Scale-free weighted gene co-expression networks were constructed using the “WGCNA” package in R [68]. The expression matrices for different kernels from XNY22 and XNY27 were planted at CHS, and YA were clustered by treatment to detect outliers. Secondly, we selected the appropriate soft threshold power β according to the pick soft threshold function in WGCNA, and we built a hierarchical clustering tree according to the calculated soft threshold to detect gene modules, and at the same time built an unsigned type TOM matrix. The minModuleSize parameter was set to 30, and the mergeCutHeight was set to 0.25. Finally, the Pearson correlation coefficient was calculated to perform principal component analysis (PCA) on all genes in each co-expression module, and the principal component 1 (PC1) was called the eigenvector (module eigengene, ME) of this module. The correlation r between ME values of each module and different traits was calculated. The closer the absolute value of the correlation between a trait and a module is to 1, the more related the genes of the module and the trait are. Through this heatmap, biologically meaningful co-expression modules can be found for subsequent analysis. In this study, the modules with |r| > 0.5 and *p* < 0.05 were set as specific modules. In a particular module, hub genes were filtered according to the criteria of |GS| > 0.2 (gene significance) and |MM| > 0.8 (module membership). According to the results of DEGs, the differential hub genes were extracted for subsequent analysis, and the pathway enrichment of co-expressed genes in their modules was paid attention.

### 2.6. GO Enrichment Analysis

Enrichment analysis was performed based on key modules and differential hub genes. GO term enrichment analysis was performed at AGRiGO (http://systemsbiology.cau.edu.cn/agriGOv2/index.php, accessed on 20 September 2022) with a threshold of FDR < 0.05.

## 3. Results

### 3.1. Cd Accumulation of Hybrid Maize Varieties XNY22 and XNY27 under the Different Cd Content Soil

In order to investigate the difference of hybrid maize varieties XNY22 and XNY27 under the different Cd content soil, we measured the Cd concentration of immature kernels, leaves, roots, and stems of 8 DAP, 24 DAP, and 40 DAP of XNY22 and XNY27 from CHS (Cd concentration 0.42 mg/kg) and YA (Cd concentration 2.73 mg/kg), by using the ZEEnit700P atomic absorption spectrometer (Analytikjena, Germany). No significant differences of Cd concentration were detected in roots of 8, 24, and 40 DAP between the CHS and YA locations (Figure 1A). In leaves, the Cd content of XNY22 was detected to be significantly higher than XNY27 of 8, 24, and 40 DAP in YA, and 8 and 24 DAP in the CHS location (Figure 1C). In stems and kernels, the Cd content of XNY22 was detected to be significantly higher than XNY27 of 8, 24, and 40 DAP in the YA location (Figure 1B,D).

### 3.2. Sequencing Quality Statistics

A total of 24 libraries of the kernels of 8 DAP and 24 DAP of XNY22 and XNY27 from CHS and YA were sequenced by using the Illumina paired-end sequencing method with three biological replicates. In total, RNA sequencing yielded 42.5 to 74.3 million high-quality clean reads, with 52.3 million clean reads per library on average. An overview of RNA-Seq data for 24 libraries is shown in Table 1. Totals of 51,752,174, 65,521,638, 49,745,935, 60,761,531, 46,435,704, 48,868,606, 45,875,188, and 49,747,555 clean reads for XNY22-YA-8, XNY22-YA-24, XNY27-YA-8, XNY27-YA-24, XNY22-CHS-8, XNY22-CHS-24, XNY27-CHS-8, and XNY27-CHS-24 were produced by RNA-Seq on an Illumina Hiseq platform, respectively. A total of 1256 billion high-quality clean reads were obtained from the 24 libraries, and the average GC content and base of each sample were 55.88%. The Q20 ranged from 95.86% to 96.38%, with an average of 96.09%. The Q30 ranged from 89.94% to 91.05%, with an average of 90.42% (Table 1). Finally, 52,569 unigenes were subsequently assembled from the 24 libraries.

Pearson correlations among the XNY22-YA-8, XNY22-YA-24, XNY27-YA-8, XNY27-YA-24, XNY22-CHS-8, XNY22-CHS-24, XNY27-CHS-8, and XNY27-CHS-24 replicates ranged from 0.73 to 1 (Appendix A). Principal component analysis results showed that PC1 and PC2 explained 84.2% and 10.8% of the gene expression variations among all samples, respectively, indicating striking differences in gene expression profiles (Appendix A).

### 3.3. DEGs Identified between High and Low Cd Accumulation in XNY22 and XNY27

In order to identify the DEGs of the kernel at 8 DAP and 24 DAP, the criteria of padj (*p* value was adjusted) < 0.05 was used. Differential expression analysis of two samples was performed using the DESeq R package. A total of 55 DEGs were identified between XNY22-CHS-8 and XNY22-YA-8, and 11 and 44 genes were upregulated and downregulated in XNY22-YA-8, respectively. A total of 23 (41.8%) DEGs were reported in Cheng’s study [53], three (5.5%) DEGs were reported in Peng’s study [54], and one was reported in these two previous studies. A total of 92 DEGs were identified between XNY22-CHS-24 and XNY22-YA-24, and 46 and 46 genes were upregulated and downregulated in XNY22-YA-24, respectively. A total of 40 (43.5%) DEGs were reported in Cheng’s study [53], eight (8.7%) DEGs were reported in Peng’s study [54], and four were reported in these two previous studies. A total of 22 DEGs were identified between XNY27-CHS-8 and XNY27-YA-8, and 8 genes were upregulated and 14 genes were downregulated in XNY27-YA-8. Five (22.7%) DEGs were reported in Cheng’s study [53], one (4.5%) DEG was reported in Peng’s study [54], and one was reported in these two studies. A total of 542 DEGs were identified between XNY27-CHS-24 and XNY27-YA-24, and 313 genes were upregulated and 229 genes were downregulated in XNY27-YA-24. A total of 124 (22.9%) DEGs were reported in Cheng’s study [53], 18 (3.3%) DEGs were reported in Peng’s study [54], and four were reported in these two studies (Figure 2A; Appendix A). Among these 542 DEGs, 9 were related to ion, metal, or heavy metal transport. In addition, *Zm00001d033564*, *Zm00001d048731*, and *Zm00001d034571* were the candidate genes of QTL mapping of root diameter, perimeter, and bushiness in maize seedling under Cd stress, respectively [48]. Furthermore, 12 DEGs were identified both in XNY22-CHS-8 vs. XNY22-YA-8 and XNY22-CHS-24 vs. XNY22-YA-24 comparison groups (Figure 2B), and seven DEGs were identified in XNY27-CHS-8 vs. XNY27-YA-8 and XNY27-CHS-24 vs. XNY27-YA-24 comparison groups (Figure 2C, Appendix A).

### 3.4. Construction of Co-Expression Network by WGCNA

To uncover important genes and regulation pathways involved in maize kernel filling period under the different Cd content soil, we performed the R WGCNA package on all 24 samples. A soft-threshold power of nine was introduced into the network topology to reveal the scale independence and mean connectivity of the network (Figure 3A,B).

### 3.5. Candidate-Hub Gene Analysis of XNY22 and XNY27

We used WGCNA to identify hub genes; it clusters gene sets with similar expression patterns into modules. Based on the fragments per kilobase per million mapped (FPKM) expression matrix and phenotypic traits, including CHS and YA locations and two developmental stages, a total of 52,569 genes in 24 samples were clustered and divided into 37 modules, which we decorated with diacritical colors (Figure 4). We explored the relationship between maize modules and Cd content in maize kernels at high and low soil Cd level.

A total of six modules, including darkorange (123 hub genes), pink (533 hub genes), blue (1224 hub genes), violet (39 hub genes), darkolivegreen (69 hub genes), and midnightblue (156 hub genes), were obtained in XNY22-YA-8. Five modules, including lightsteelblue1 (48 hub genes), plum1 (47 hub genes), black (472 hub genes), paleturquoise (75 hub genes), and brown (817 hub genes), were obtained in XNY27-YA-8. Four modules, including darkorange2 (30 hub genes), salmon (315 hub genes), green (654 hub genes), and saddlebrown (115 hub genes), were obtained in XNY22-CHS-8. Seven modules, including white (143 hub genes), pink (601 hub genes), blue (1646 hub genes), green (891 hub genes), saddlebrown (120 hub genes), black (739 hub genes), and yellow (663 hub genes), were obtained in XNY27-CHS-8. One module, orangered4 (79 hub genes), was obtained in XNY22-YA-24. Two modules, including greenyellow (307 hub genes) and ivory (24 hub genes), were obtained in XNY27-YA-24. Eight modules, including darkgreen (116 hub genes), royablue (135 hub genes), cyan (197 hub genes), lightgreen (129 hub genes), skyblue3 (47 hub genes), ivory (28 hub genes), plum2 (15 hub genes), and red (342 hub genes), were obtained in XNY22-CHS-8. Four modules, including ivory (42 hub genes), lightcyan1 (90 hub genes), lightcyan (199 hub genes), and darkturquoise (116 hub genes), were obtained in XNY27-CHS-24 (Figure 5). These modules were positively or negatively correlated with different soil Cd content, and the genes in the corresponding modules were upregulated or downregulated, indicating that the genes respond to Cd stress at 8 and 24 DAP of XNY22 and XNY27.

### 3.6. GO Enrichment Analysis of Candidate Hub Genes

Combining the hub genes and DEGs, we obtained 71 candidate genes which were detected in hub genes and DEGs. The 71 candidate hub genes included greenyellow (56 hub genes) and ivory (14 hub genes) in XNY27-YA-24, and pink (1 hub gene) in XNY22-YA-8 (Appendix A). To further understand the biological functions of genes on the Cd content in maize kernel related module, we performed GO enrichment analysis of genes in the greenyellow and ivory modules in XNY27-YA-24. Genes in the greenyellow module were grouped into 41 significant enrichment GO terms, including 38 biological process terms and 3 cellular components related terms (Figure 6A). Genes in the biological process terms were primarily matched and classified into response to inorganic substance, abscisic acid, arsenic-containing substance, hormone, metal ion, oxidative stress, and so on. Genes in the cellular component category included the lipid particle, monolayer-surrounded lipid storage body, and endoplasmic reticulum (Figure 6A, Appendix A). Genes in the ivory module were grouped into 15 significant enrichment GO terms, including four biological process terms, seven molecular functions, and four cellular components related terms (Figure 6B). Genes in the biological process category were matched and classified into negative regulation of hydrolase activity, regulation of hydrolase activity, negative regulation of molecular function, and negative regulation of catalytic activity. In the molecular function term, most of the genes were classified into oxidoreductase activity, carbon-oxygen lyase activity, monooxygenase activity and enzyme inhibitor activity. The most abundant GO terms in the cellular component category included the integral to membrane, intrinsic to membrane, extracellular region, and so on (Figure 6B, Appendix A). Thus, we speculated that these pathways may play an important role in regulating the Cd content in the maize kernels. Thereafter, we found that the GO term, GO: 0010038 (response to metal ion) included 56 genes in the greenyellow module, and five genes (*Zm00001d039933*, *Zm00001d012764*, *Zm00001d027740*, *Zm00001d020888*, and *Zm00001d048073*) were DEGs. Moreover, gene connectivity analyses were performed on the genes in the greenyellow module. *Zm00001d039933*, *Zm00001d012764*, *Zm00001d027740*, *Zm00001d020888*, and *Zm00001d048073* showed higher connectivity (Figure 7, Appendix A).

### 3.7. Candidate Hub Genes Network Interaction in Phenotypic Significant Enrichment Module

Here, gene network visualization and gene connectivity analysis were performed on the 71 candidate hub genes, which were significantly related with Cd content of maize kernel at high and low soil Cd level. The top 20 genes with highest connectivity in 71 candidate hub genes were used to map the gene co-expression network (Figure 8, Appendix A). A total of 193 interacting genes, including 71 candidate hub genes, were extracted from the co-expression network (Figure 8). Subsequently, the GO enrichment analysis of these interacting genes was performed, and the results showed that the responses to inorganic substance, metal ion, abiotic stimulus, hydrogen peroxide, oxidative stress, stimulus, and other processes were enrichment (Appendix A). Moreover, these five genes, related above (*Zm00001d039933*, *Zm00001d012764*, *Zm00001d027740*, *Zm00001d020888*, and *Zm00001d048073*), were enriched in GO term response to metal ion. Therefore, we speculate that genes are related to the regulation of Cd content in maize kernels.

## 4. Discussion

Heavy metals, including Cd, lead, and arsenic, are important abiotic factors inhibiting plant growth, decreasing food production, potential threating food safety, and directly affecting human health [69,70]. Therefore, it is important to extend our understanding of how plants absorb and translocate these heavy metals and their defenses to these stress. Until now, many previous studies have been conducted on Cd absorption, translocation, chelation, and detoxification in model species, such as rice [71,72,73] and Arabidopsis [74].

Maize is a hyperaccumulator of Cd, and it is an important crop to decrease Cd availability in soil. The Cd content of different maize organs is different [75,76]. Usually, the Cd content in the maize kernel is lower than in other organs. Transcriptomics was conducted on maize seedling roots in several previous studies, but few in maize kernel. In this study, transcriptomics was performed in maize kernel. Differential gene expressions induced by Cd stress were analyzed using comparative RNA-Seq technology to compare gene expression profiles of hybrid maize varieties XNY22 and XNY27 kernel under high and low Cd concentration soil. In total, 55, 92, 22, and 542 DEGs were identified between XNY22-CHS-8 vs. XNY22-YA-8, XNY22-CHS-24 vs. XNY22-YA-24, XNY27-CHS-8 vs. XNY27-YA-8, and XNY27-CHS-24 vs. XNY27-YA-24, respectively (Figure 2, Appendix A).

In this study, no significant differences of Cd concentration were detected in roots between CHS and YA, and the Cd content in stems, leaves, and kernels of XNY22 were significantly higher than XNY27 at YA (Figure 1). These results implied that the transporter of Cd participated in this process. The DEGs annotations showed that nine were related to ion, metal, or heavy metal transport. This suggests that these nine genes may be related with Cd transport in maize.

Nowadays, few genes in maize-related Cd response are cloned, such as *ZmHMA3* [49,51], *ZmHMA2* [58], *ZmUBP15*, *ZmUBP16*, and *ZmUBP19* [59]. One possible reason is that a few major genes control the Cd uptake and translocation in maize. Therefore, more studies about Cd response are needed. The previous studies of maize roots transcriptome showed that the GO enrichment analyses of DEGs were mainly involved in nucleic acid binding transcription factor activity, response to stress, oxidoreductase activity and iron ion binding [54], DNA binding transcription factor, S-adenosylmethionine biosynthesis, response to oxidative stress, biotic stimulus, and peroxidase activity [52]. In the current study, the WGCNA categorized the 9599 Cd stress-responsive genes into 37 modules at 8 and 24 DAP immature kernels of XNY22 and XNY27 (Figure 4). Combining these hub genes and the DEGs, 71 candidate hub genes were detected, which included greenyellow (56 hub genes), ivory (14 hub genes), and pink (1 hub gene) (Appendix A). GO analysis of candidate hub genes in the greenyellow module were mainly involved in response to inorganic substance, abscisic acid, arsenic-containing substance, hormone, metal ion, oxidative stress, and so on, and genes in the ivory module were mainly involved in negative regulation of hydrolase activity, regulation of hydrolase activity, negative regulation of molecular function, negative regulation of catalytic activity, and so on (Figure 6, Appendix A). The GO term, GO: 0010038 (response to metal ion), included 56 genes in the greenyellow module, and *Zm00001d039933*, *Zm00001d012764*, *Zm00001d027740*, *Zm00001d020888*, and *Zm00001d048073* were DEGs in XNY27-CHS-24 vs. XNY27-YA-24 (Figure 7, Appendix A). There results imply that these five genes may participate in maize kernel response to Cd stress.

Meanwhile, based on the 71 candidate hub genes, the gene connectivity analysis showed 193 interacting genes for the top 20 genes with the highest connectivity in 71 candidate hub genes. GO enrichment analysis of these interacting genes showed that the responses to inorganic substance, metal ion, abiotic stimulus, hydrogen peroxide, oxidative stress, stimulus, and other processes were enrichment (Appendix A). It is different that the underlying mechanism of seedling roots and kernels is responsive to Cd stress.

## 5. Conclusions

In summary, a large number of maize kernel DEGs in response to Cd stress were identified. The WGCNA categorized the 9599 Cd stress-responsive hub genes into 37 different gene network modules. We obtained 71 candidate genes which were both detected in hub genes and DEGs. GO enrichment analysis of genes in the greenyellow module in XNY27-YA-24 showed that the responses to metal ion, inorganic substance, abiotic stimulus, hydrogen peroxide, oxidative stress, stimulus, and other processes were enrichment. RNA-Seq combined with the WGCNA analysis revealed the regulatory network of hub genes in significant modules. These results provide a foundation for further study of maize response to Cd stress. The putative key genes and pathways identified may be used to explore the progress of Cd absorption, transfer, and accumulation in maize. In addition, this dataset will be useful for unraveling the underlying mechanism of Cd accumulation in maize kernel.

## Figures and Tables

**Figure 1 genes-13-02130-f001:**
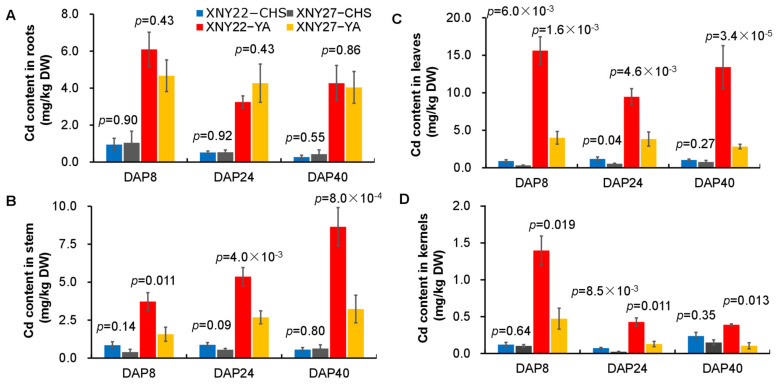
Cd content in roots (**A**), stems (**B**), leaves (**C**), and kernels (**D**) of hybrid maize XNY22 and XNY27 at 8, 24, and 40 DAP in the CHS and YA locations.

**Figure 2 genes-13-02130-f002:**
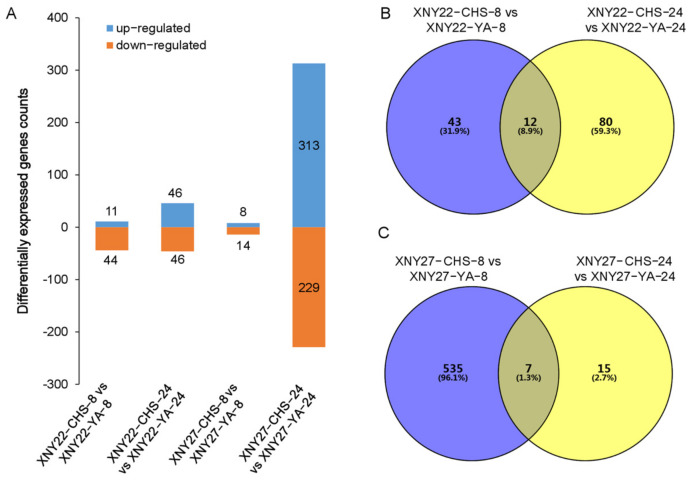
The differentially expressed genes analysis. (**A**) The number of DEGs in four groups. (**B**) Venn diagram of hybrid maize XNY22 DEGs in two groups. (**C**) Venn diagram of hybrid maize XNY27 DEGs in two groups.

**Figure 3 genes-13-02130-f003:**
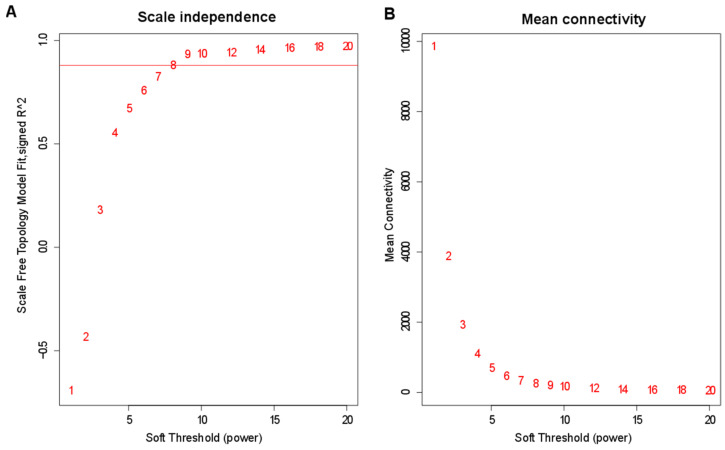
Module identification by weighted gene co-expression network analysis (WGCNA). (**A**,**B**) represent the soft threshold with scale independence and mean connectivity.

**Figure 4 genes-13-02130-f004:**
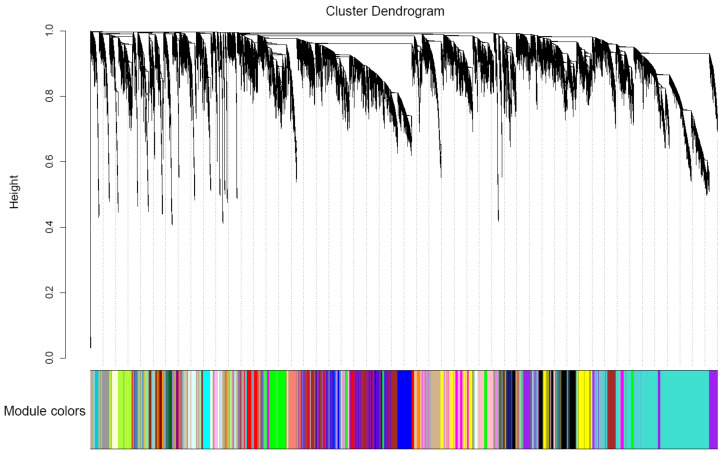
Hierarchical dendrogram reveals co-expression modules identified by WGCNA. A total of 37 modules were identified based on calculation of eigengenes; each module was decorated with a different color.

**Figure 5 genes-13-02130-f005:**
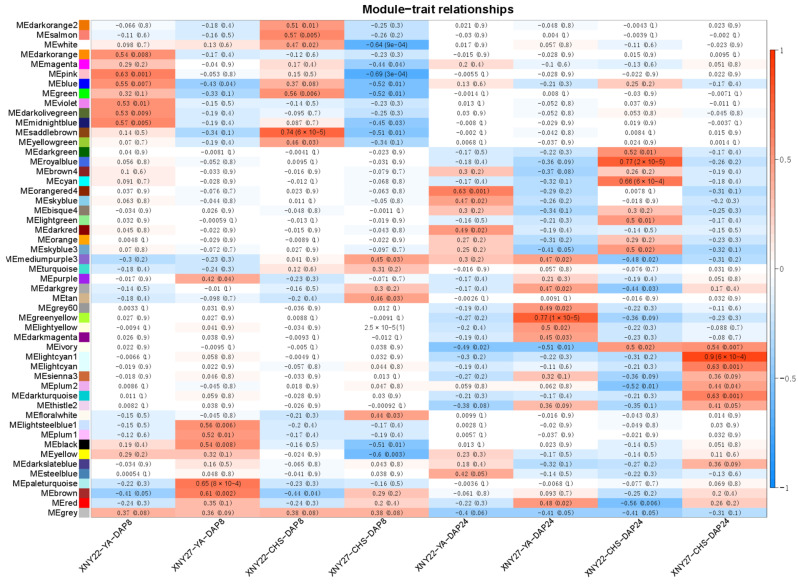
The correlation coefficient and correlation significance between the module and different kernel developmental stage hybrid of Cd stress. Each row in the table corresponds to a consensus module, and each column to different location, kernel developmental stage, and hybrid. The module name is shown on the *y*-axis, and the time point is shown on the *x*-axis.

**Figure 6 genes-13-02130-f006:**
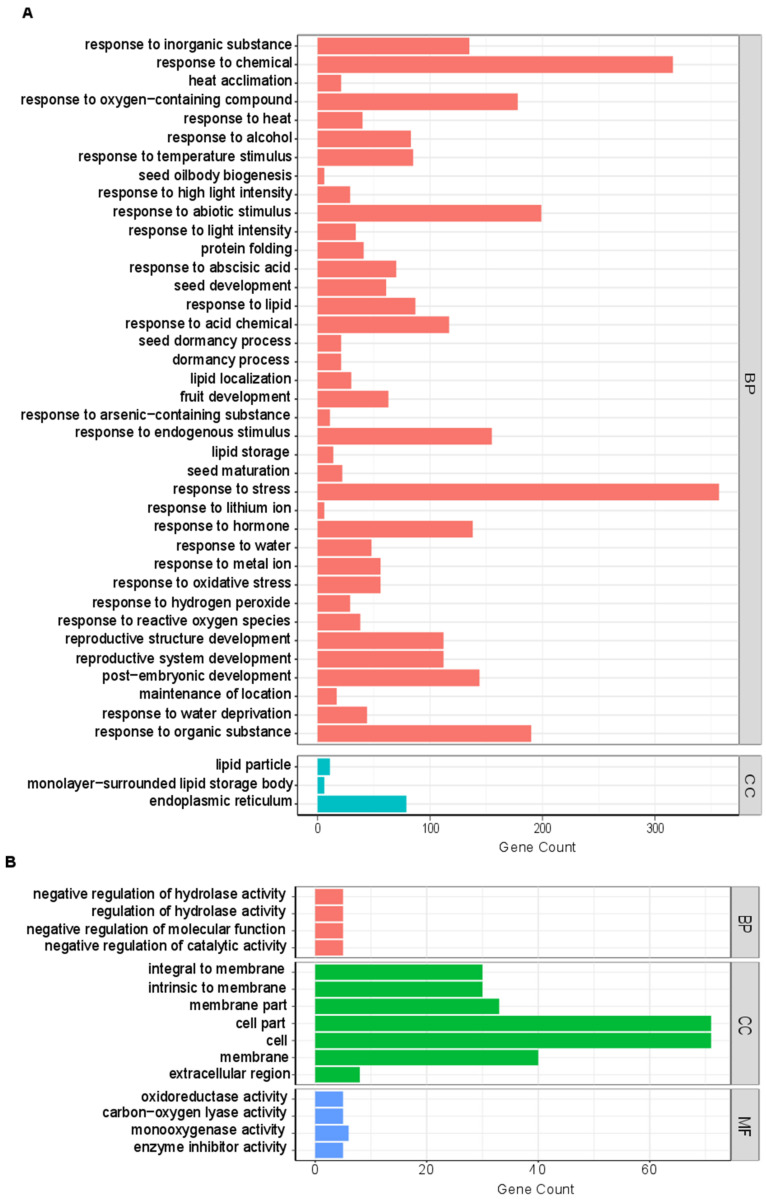
GO term enrichment of genes in the greenyellow (**A**) and ivory (**B**).

**Figure 7 genes-13-02130-f007:**
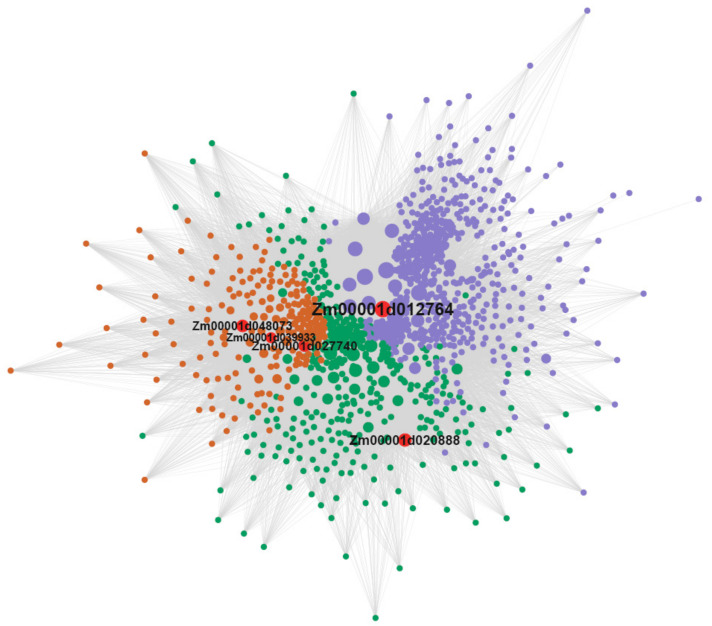
Analysis of candidate hub genes network interaction in greenyellow module.

**Figure 8 genes-13-02130-f008:**
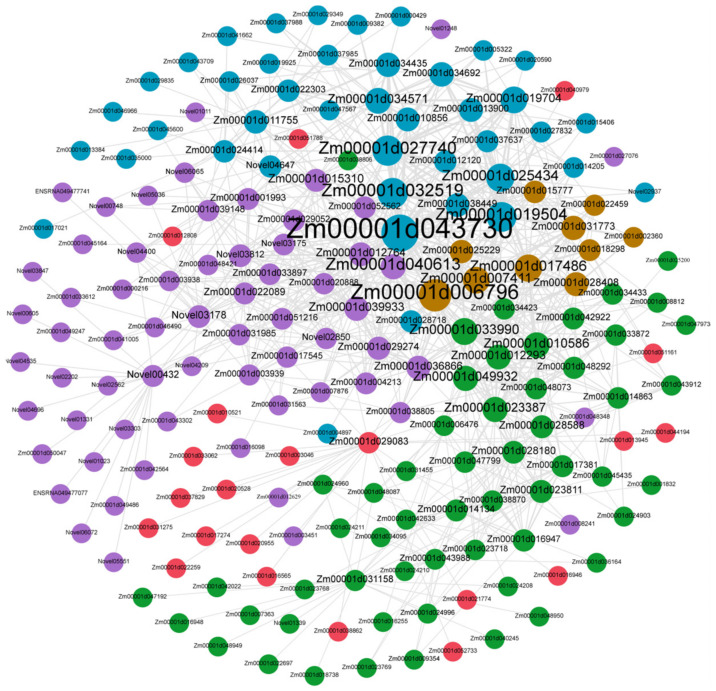
Analysis of candidate hub genes network interaction in the phenotypic significant enrichment module. The correlation network visualization of the interactions showed the connection between the top 20 genes with the highest connectivity in 71 candidate hub genes.

**Table 1 genes-13-02130-t001:** Statistical analyses and mapping results of RNA sequencing reads.

Sample Name	Total Raw Reads	Total Clean Reads	Clean Bases	Q20	Q30	GC Content
XNY22-YA-8-1	61,002,492	58,995,782	4.42G	95.89%	90.00%	55.10%
XNY22-YA-8-2	49,000,066	47,314,434	3.55G	95.86%	89.94%	55.16%
XNY22-YA-8-3	50,773,422	48,946,308	3.67G	95.90%	90.03%	55.30%
XNY22-YA-24-1	57,585,600	55,670,330	4.18G	95.94%	90.06%	57.05%
XNY22-YA-24-2	71,045,424	68,906,076	5.17G	96.11%	90.55%	55.47%
XNY22-YA-24-3	74,261,462	71,988,508	5.4G	96.38%	91.05%	56.07%
XNY27-YA-8-1	49,047,692	47,515,340	3.56G	96.21%	90.64%	54.99%
XNY27-YA-8-2	54,774,602	53,113,822	3.98G	96.06%	90.31%	55.35%
XNY27-YA-8-3	50,077,846	48,608,644	3.65G	96.05%	90.29%	54.80%
XNY27-YA-24-1	65,085,908	62,577,096	4.69G	96.30%	90.86%	56.86%
XNY27-YA-24-2	61,293,978	59,407,268	4.46G	96.26%	90.83%	56.71%
XNY27-YA-24-3	62,471,238	60,300,228	4.52G	96.17%	90.63%	57.02%
XNY22-CHS-8-1	45,920,496	44,668,098	3.35G	96.16%	90.61%	55.41%
XNY22-CHS-8-2	50,754,324	49,405,462	3.71G	96.35%	91.00%	55.62%
XNY22-CHS-8-3	46,733,440	45,233,552	3.39G	96.18%	90.64%	55.65%
XNY22-CHS-24-1	51,689,096	49,994,612	3.75G	96.13%	90.45%	55.07%
XNY22-CHS-24-2	44,128,234	42,698,694	3.2G	96.09%	90.35%	56.72%
XNY22-CHS-24-3	55,802,774	53,912,512	4.04G	96.03%	90.22%	56.36%
XNY27-CHS-8-1	45,568,500	44,342,446	3.33G	96.13%	90.52%	54.78%
XNY27-CHS-8-2	47,235,914	45,699,000	3.43G	95.95%	90.06%	55.69%
XNY27-CHS-8-3	49,151,816	47,584,118	3.57G	96.03%	90.22%	56.59%
XNY27-CHS-24-1	52,646,388	50,889,064	3.82G	96.14%	90.47%	56.59%
XNY27-CHS-24-2	57,725,532	55,861,930	4.19G	96.07%	90.31%	57.71%
XNY27-CHS-24-3	44,060,200	42,491,670	3.19G	95.95%	90.09%	55.09%

XNY22-YA-8–XNY27-CHS-24 represent that the three repeat samples were collected from immature kernels of 8 DAP and 24 DAP of XNY22 and XNY27 from two environments, YA and CHS.

## Data Availability

The data presented in this study are available upon request from the corresponding author.

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
