# Peer review of "WGCNA Analysis Revealed the Hub Genes Related to Soil Cadmium Stress in Maize Kernel (*Zea mays* L.)"

_genes, 2022, doi:10.3390/genes13112130_

Round 1
Reviewer 1 Report
The manuscript entitled “WGCNA Analysis Revealed the Hub Genes Related to Soil Cadmium Stress in Maize Kernel (Zea mays L.)” is well organised. Authors have screened several genes to show the effect of Cd in different maize hybrids. However, a few major necessary corrections are needed to incorporate into the manuscript.
In figure 6A., the authors suggested to separate a group for transporters genes. It will give an idea to the readers of the uptake mechanism in maize hybrids. It will be better to split a graph to explain the different groups of transporters. Figure 1. indicates that the different hybrids affect the accumulation of Cd. So, it will be better to show the transporter's genes. Also, explain it in the result and discussion sections.
Author Response
Thank you for the very nice comments. Based on the GO of DEGs, we didn’t found the transporters GO item was enrichment. But some genes were related with metal or heavy metal ion transporter based on the annotation. This part was added in the revised manuscript.
Reviewer 2 Report
The authors of this article screened the transcriptome of two maize genotypes in relation to cadmium accumulation. They performed WGCNA analysis to explore highly correlated modules and co-expressed genes. They identified 37 different gene network modules from the cadmium stress-responsive hub genes and thus obtained 71 candidate genes from this analysis.
Considering the toxicity of cadmium and its effect on humans and animals, its important to understand the molecular mechanism of cadmium accumulation to be able to breed lines with low cadmium. However, I have a couple of questions/suggestions:
What software was used for filtering the raw reads? cite it
I suggest you create and mention a GitHub repo or any other repository or supplementary files with the line of R codes with the input dataset used in the WGCNA analysis and other related analyses to ensure repeatability.
Several transcriptomic studies have been performed for the maize root response to cadmium stress in the past. It will be beneficial if a thorough comparison of the previous findings of these studies is done with the current study. Also, in a recent publication (https://academic.oup.com/genetics/article/218/3/iyab087/6294935?login=true) numerous phenotypes are analyzed to detect pleiotropic loci. Are any of the genes identified in the current study overlapping with the peaks identified in the mentioned publication? will it be possible to find a phenotype associated with cadmium accumulation to enable indirect selection for the trait aiding breeding of maize lines with less cadmium accumulation?
The text in figure 5 is not readable, please replot it to ensure the readability of the numbers and the axis text.
Lastly, there are numerous grammatical errors throughout the manuscript that need be addressed.
Author Response
Thank you for the very nice comments.
Q1: What software was used for filtering the raw reads? cite it
A1: Trimmomatic was used for filtering the raw reads. The reference was cited in the revised manuscript.
Q2: I suggest you create and mention a GitHub repo or any other repository or supplementary files with the line of R codes with the input dataset used in the WGCNA analysis and other related analyses to ensure repeatability.
A2: we created a GitHub, and input the R codes in this link.
Q3: Several transcriptomic studies have been performed for the maize root response to cadmium stress in the past. It will be beneficial if a thorough comparison of the previous findings of these studies is done with the current study. Also, in a recent publication (https://academic.oup.com/genetics/article/218/3/iyab087/6294935?login=true) numerous phenotypes are analyzed to detect pleiotropic loci. Are any of the genes identified in the current study overlapping with the peaks identified in the mentioned publication? Will it be possible to find a phenotype associated with cadmium accumulation to enable indirect selection for the trait aiding breeding of maize lines with less cadmium accumulation?
A3: Thank you for the suggestions. We compared the DEGs with the previous studies, and found several DEGs were detected in current and previous studies. Moreover, three DEGs were found that overlapped with the QTLs of bushiness, perimeter and diameter of root in maize seedling under Cd stress.
Q4: The text in figure 5 is not readable, please replot it to ensure the readability of the numbers and the axis text.
A4: We have modified it in the revised manuscript.
Q5: Lastly, there are numerous grammatical errors throughout the manuscript that need be addressed.
A5: We have modified it in the revised manuscript.

Round 2
Reviewer 1 Report
Authors have incorporated all the suggestions perfectly. I recommend manuscript to published in revised version.
Author Response
Thank you for the very nice comments.